# Abscisic Acid Affects Phenolic Acid Content to Increase Tolerance to UV-B Stress in *Rhododendron chrysanthum* Pall.

**DOI:** 10.3390/ijms25021234

**Published:** 2024-01-19

**Authors:** Xiangru Zhou, Fushuai Gong, Jiawei Dong, Xiaoru Lin, Kun Cao, Hongwei Xu, Xiaofu Zhou

**Affiliations:** Jilin Provincial Key Laboratory of Plant Resource Science and Green Production, Jilin Normal University, Siping 136000, China

**Keywords:** *Rhododendron chrysanthum* Pall., metabolome, transcriptome, UV-B stress, phenolic acid, abscisic acid

## Abstract

The presence of the ozone hole increases the amount of UV radiation reaching a plant’s surface, and UV-B radiation is an abiotic stress capable of affecting plant growth. *Rhododendron chrysanthum* Pall. (*R. chrysanthum*) grows in alpine regions, where strong UV-B radiation is present, and has been able to adapt to strong UV-B radiation over a long period of evolution. We investigated the response of *R. chrysanthum* leaves to UV-B radiation using widely targeted metabolomics and transcriptomics. Although phytohormones have been studied for many years in plant growth and development and adaptation to environmental stresses, this paper is innovative in terms of the species studied and the methods used. Using unique species and the latest research methods, this paper was able to add information to this topic for the species *R. chrysanthum*. We treated *R. chrysanthum* grown in a simulated alpine environment, with group M receiving no UV-B radiation and groups N and Q (externally applied abscisic acid treatment) receiving UV-B radiation for 2 days (8 h per day). The results of the MN group showed significant changes in phenolic acid accumulation and differential expression of genes related to phenolic acid synthesis in leaves of *R. chrysanthum* after UV-B radiation. We combined transcriptomics and metabolomics data to map the metabolic regulatory network of phenolic acids under UV-B stress in order to investigate the response of such secondary metabolites to stress. L-phenylalanine, L-tyrosine and phenylpyruvic acid contents in *R. chrysanthum* were significantly increased after UV-B radiation. Simultaneously, the levels of 3-hydroxyphenylacetic acid, 2-phenylethanol, anthranilate, 2-hydroxycinnamic acid, 3-hydroxycinnamic acid, α-hydroxycinnamic acid and 2-hydroxy-3-phenylpropanoic acid in this pathway were elevated in response to UV-B stress. In contrast, the study in the NQ group found that externally applied abscisic acid (ABA) in *R. chrysanthum* had greater tolerance to UV-B radiation, and phenolic acid accumulation under the influence of ABA also showed greater differences. The contents of 2-phenylethanol, 1-o-p-coumaroyl-β-d-glucose, 2-hydroxy-3-phenylpropanoic acid, 3-(4-hydroxyphenyl)-propionic acid and 3-o-feruloylquinic ac-id-o-glucoside were significantly elevated in *R. chrysanthum* after external application of ABA to protect against UV-B stress. Taken together, these studies of the three groups indicated that ABA can influence phenolic acid production to promote the response of *R. chrysanthum* to UV-B stress, which provided a theoretical reference for the study of its complex molecular regulatory mechanism.

## 1. Introduction

Ultraviolet radiation is a component of solar radiation that can affect plant growth and development [1]. Due to the ozone hole, the UV-B radiation reaching the earth exceeds the range acceptable to plants and affects plant health [2]. UV-B radiation can induce DNA damage, produce reactive oxygen species (ROS) and impair photosynthesis. The response of plants to UV-B light is regulated by the wavelength, intensity and duration of UV-B radiation [3]. Plants take UV-B (280–315 nm) as an environmental signal and a potential abiotic stressor affecting plant development and adaptation [4]. Previous studies have shown that exposing rosemary plants to excessive UV-B radiation can increase their total phenolic content [5]. Under normal conditions, the content of brachybrachine (an indole monoterpenoid alkaloid) in brachybrachine is very low, but after UV-B irradiation, the content of this alkaloid will reach 1.8% of the dry weight of the leaves of these plants [6]. Low doses of UV-B radiation will lead to the accumulation of *Arabidopsis* flavonoids [7]. Selenium can improve plant tolerance to stress, while UV-B enhancement can affect the yield and quality of winter wheat [8]. These studies indicate that UV-B has a significant impact on all aspects of the whole growth and development process of plants, which is worthy of further study.

*Rhododendron chrysanthum* Pall. (*R. chrysanthum*), commonly known as cowhide tea, is born in alpine grassland areas or on mossy strata at altitudes of 1000–2506 m above sea level, and it originates from southeastern Jilin, China. Plants cannot move to avoid the effects of environmental stresses when faced with complex and variable living environments, so plants have developed unique adaptations and defense mechanisms to avoid damage over time. Growing at high altitude, cold and exposed to UV-B radiation all year round, *R. chrysanthum* has developed unique adaptive mechanisms over a long period of evolution. It has been shown that UV-B stress regulates the response of *R. chrysanthum* to stress by altering primary metabolites (mainly involving amino acids and carbohydrates) and changes in related genes [9]. Acetylation proteomics studies of *R. chrysanthum* after UV-B stress suggest that UV-B stress initiates the photosynthesis pathway and that PSII proteins may be modified by acetylation to mitigate the damage caused by the stress [10]. Changes in secondary metabolites after UV-B stress in *R. chrysanthum* have been less studied, and further research is needed.

Phenolic acid is a secondary metabolite with a wide range of biological activities, such as antimicrobial, antioxidant, antiviral, and anticancer, and may be an important metabolite for clinical applications [11]. Phenolic acids are substances containing phenolic rings and organic carboxylic acid functional groups. Phenolic rings are able to confer antioxidant properties to phenolic acids, which are important antioxidant compounds in rice [12]. Some studies have shown that marigold extract phenolic acids have significant antioxidant effects and can be used as a new source of natural antioxidants [13]. Investigating the changes in phenolic acids in *R. chrysanthum* after UV-B stress may play an important role in revealing the mechanism of resistance to UV-B stress in *R. chrysanthum*.

Phytohormones are signaling compounds that regulate plant growth and development and play important roles in response to environmental stress [14]. ABA coordinates plant responses to adversity, regulates complex metabolic and physiological mechanisms and is essential for survival in a changing environment [15]. ABA is recognized as a common mediator that controls the adaptive response of plants to various environmental stresses [16]. Endogenous ABA levels increase in plants such as maize under drought stress [17]. Abscisic acid acts mainly on stomata and is able to improve drought tolerance in tomato by reducing water loss and, thus, delaying dehydration and hydraulic dysfunction [18]. Tobacco cells have elevated levels of endogenous ABA following salt stress [19]. The application of exogenous ABA to plants enhances their adaptive response to abiotic stresses [20]. External application of ABA can improve alfalfa’s frost tolerance to some extent [21]. How well the externally applied ABA adapted *R. chrysanthum* after UV-B stress was an important part of this study.

The study of transcriptomics can provide genome-wide information on gene structure and gene function, which is conducive to unraveling the molecular mechanism of UV-B stress resistance in *R. chrysanthum* [22]. Since there is no reference genome for *R. chrysanthum*, transcriptomics can be used to assess its transcriptional activity and understand the expression of genes in its secondary metabolic pathways [23]. The establishment of the transcriptome profile of *Polygonum minus* facilitates future studies on its effects on environmental stress and the biosynthesis of secondary metabolites, and will contribute to the further development of the genetic resources of this herbaceous plant [24]. Widely targeted metabolomics analysis based on liquid chromatography–tandem mass spectrometry (LC-MS/MS) is a reliable method capable of detecting a wide range of plant metabolites [25]. In order to better understand the changes produced in the leaves of *R. chrysanthum* after UV-B irradiation, we identified and quantified metabolites in the leaves of *R. chrysanthum* before and after treatment using UPLC-MS/MS analysis. Substances identified include primary metabolites (amino acids, etc.) and secondary metabolites (phenolic acids, terpenoids, etc.). The integrated analysis of transcriptomics and metabolomics can lead to more accurate and informative results from histological experiments, increasing our understanding of metabolic networks and a more complete view of molecular pathways [26]. Therefore, we analyzed metabolomics in conjunction with transcriptomics to explore the possible causes of UV-B radiation resistance in the *R. chrysanthum*.

In this study, the metabolic pathways of *R. chrysanthum* leaves under UV-B stress were investigated using a combination of widely targeted metabolomics and transcriptomics, using state-of-the-art biotechnology and research methods. As a species growing in a unique environment, adaptive changes during long-term stress can enrich the theory of UV-B radiation resistance in *R. chrysanthum*. Although there are many studies on ABA, the role of ABA in the UV-B stress resistance of *R. chrysanthum* is less studied. In this paper, we investigated the changes in phenolic acid compounds in ABA-treated *R. chrysanthum*, which enriched the knowledge of ABA in plant stress resistance, which is helpful to understand the important reasons of *R. chrysanthum*’s resistance to UV-B radiation and is beneficial in enriching the theory of plant stress resistance.

## 2. Results

### 2.1. Large Amounts of Secondary Metabolites Were Produced in the Leaves of R. chrysanthum after UV-B Stress

The samples were identified using widely targeting metabolomics, and a total of 2148 metabolites were identified in this experiment. The 2148 metabolites can be classified into 13 categories, which are flavonoids (487), phenolic acids (394), terpenoids (227), amino acids and derivatives (188), lipids (134), alkaloids (117), organic acids (108), lignans and coumarins (91), nucleotides and derivatives (75), tannins (37), quinones (17), steroids (2), and others (271). Of the 2148 metabolites identified, 505 were primary metabolites, 1372 were secondary metabolites, and 271 were others. The statistical information of metabolites is shown in Table 1.

### 2.2. Significant Changes in Phenolic Acids in Leaves after UV-B Stress

Principal component analysis (PCA) was performed on the MN group samples to understand the overall metabolic differences between the samples and the magnitude of variability between the samples within the group. The first three principal components explained 39.01%, 22.22%, and 15.39% of the data set, respectively. According to the PCA results, the MN group samples can be well separated and have small intra-group differences (Appendix A). Screening for differential metabolites was performed using FC ≥ 1.5, FC ≤ 0.67, and VIP > 1. The results are shown in Appendix A, with 355 metabolites increased and 167 metabolites decreased between the MN groups. KEGG enrichment analysis of the differential metabolites revealed that the first few entries were “ABC transporters”, “Biosynthesis of amino acids”, “D-Amino acid metabolism”, and “Phenylalanine metabolism” (Appendix A). A count of the types and numbers of metabolites that differed between the MN groups revealed that 83 flavonoids and 79 phenolic acids were increased in secondary metabolites, and 36 phenolic acids and 30 flavonoids were decreased (Appendix A). It can be seen that the phenolic acids of *R. chrysanthum* changed significantly after UV-B stress, which may facilitate its adaptation to an adverse environment. In order to study the accumulation of phenolic acids in *R. chrysanthum* after UV-B stress, we performed a principal component analysis of the 394 phenolic acids identified and found that the MN group was able to segregate significantly, with 40.49% for PC1 and 24.01% for PC2 (Figure 1a). Of the 394 phenolic acids, 79 were increased and 36 decreased after UV-B radiation. The top-three phenolic acids that were significantly increased were leucosceptoside A, 4-o-digalloyl-3,5-di-o-galloylquinic acid, and 3-o-feruloylquinic acid-o-glucoside. The top-three phenolic acids that were significantly decreased were 2,4,6-tri-o-galloyl-d-glucose, 2,3-di-o-galloyl-β-d-glucose*, and raspberryketone glucoside (Figure 1b). Figure 1c illustrates the quantitative statistics of all metabolites and phenolic acids identified between MN groups. There were 115 phenolic acids out of 522 differential metabolites, accounting for 22%, suggesting that phenolic acids are important metabolites produced by *R. chrysanthum* leaves against UV-B radiation. KEGG enrichment analysis of phenolic acids between MN groups revealed that phenolic acids were significantly enriched in the phenylalanine metabolism and phenylpropanoid biosynthesis pathway (Figure 1d). Figure 1e shows a heatmap of the content of 115 differential phenolic acids between MN groups, and Figure 1f shows a heatmap of 27 differential phenolic acids between MN groups that were able to be annotated to the KEGG database. Figure 1g shows detailed information of the 27 differential phenolic acids. The relevant metabolite codes and corresponding name information involved in the images and tables are shown in Appendix A.

### 2.3. Significant Changes in Genes Related to Phenolic Acid Synthesis in Leaves after UV-B Stress

As the MN intergroup differential metabolites were significantly enriched in the phenylalanine metabolism pathway, we investigated the expression of genes in this pathway. Of the 672 genes annotated in this pathway, 140 were able to be expressed, and 8 were DEGs in the MN group (Figure 2a). We performed GO, KEGG classification and enrichment analysis of 672 genes and found that genes annotated for biological processes focused on cellular and metabolic processes, genes annotated for cellular components focused on cellular anatomical entity and genes annotated for molecular functions focused on binding and catalytic activity (Figure 2b). The GO enrichment results showed that 672 genes were significantly enriched to entries, such as “cellular amino acid metabolic process”, “oxidoreductase activity, acting on the CH-OH group of donors, NAD or NADP as acceptor”, “L-aspartate:2-oxoglutarate aminotransferase activity”, “pyridoxal phosphate binding”, “NAD binding” and “phenethylamine:oxygen oxidoreductase (deaminating) activity”, demonstrating the possible functions of these genes (Figure 2c). KEGG classification results showed that genes were mainly enriched in metabolic entries and were enriched in “Global and overview maps”, “Amino acid metabolism” and “Biosynthesis of other secondary metabolites” (Figure 2d). KEGG is enriched in “Phenylalanine metabolism”, “Tyrosine metabolism”, “Tropane, piperidine and pyridine alkaloid biosynthesis”, “Phenylalanine, tyrosine and tryptophan biosynthesis” and other entries (Figure 2e). We performed a heatmap count of 140 genes expressed in the ko00360 pathway, corresponding to 14 structural genes (*PAL*, *GOT1*, *GOT2*, *ASP5*, *PAT*, *TAT*, *hisC*, *MIF*, *HPPR*, K01426, *HPD*, *DDC*, *AOC3* and *PAAS*) and 11 enzymes (“4.3.1.24”, “2.6.1.1”, “2.6.1.5”, “2.6.1.9”, “5.3.2.1 “, “1.1.1.237”, “3.5.1.4”, “1.13.11.27”, “4.1.1.28”, “1.4.3.21” and “4.1.1.109”) (Figure 2f). Among these 140 genes, 8 DEGs were directly related to phenolic acid synthesis, and “TRINITY_DN104_c0_g1_i10-C_1” and “TRINITY_DN104_c0_g1_i2-C_1” were significantly downregulated; the other 6 genes were significantly upregulated after UV-B radiation (Figure 2g). We show the expression of these eight DEGs in the MN group in a bar graph (Figure 2h).

### 2.4. Strong Correlation between Phenolic Acid and Abscisic Acid in Leaves of R. chrysanthum

Abscisic acid is a phytohormone that can play a role in abiotic stress in plants, whereas L-phenylalanine is a precursor substance for phenolic acid biosynthesis. The relationship between L-phenylalanine and abscisic acid synthesis is illustrated in Figure 3a, where the content of intermediates may affect the accumulation of abscisic acid and L-phenylalanine. D-erythrose 4-phosphate and D-glyceraldehyde 3-phosphate are capable of interconverting each other, whereas D-erythrose 4-phosphate can form L-Phenylalanine through a series of reactions, and D-glyceraldehyde 3-phosphate can generate 1-deoxy-d-xylulose 5-phosphate, which, in turn, generates abscisic acid through the MEP/DOXP pathway, and, thus, abscisic acid may affect the synthesis of phenolic acids (Figure 3a). The ABA signaling process is shown in Figure 3b, where the ABA receptor receives signals from abscisic acid molecules and then metabolizes and inhibits the activity of PP2C, thereby attenuating or eliminating the inhibition of SnRK2 by PP2C and enhancing the phosphorylation of substrate levels by the SnRK2 kinase to regulate the overall response to abscisic acid in plants. The bar graphs in the figure show the expression of relevant genes. ABA controls plant response to abiotic stresses by influencing transcriptional and post-transcriptional modifications of downstream regulators in the signaling pathway, thereby enhancing plant resistance. In order to study the relationship between phenolic acids and abscisic acid, we performed a correlation analysis, and abscisic acid was negatively correlated with 13 phenolic acids and positively correlated with 14 phenolic acids. The strongest correlation was between abscisic acid and syringin, followed by picein, and all were positively correlated (Figure 3c) (Appendix A).

### 2.5. Integrating Transcriptomic and Metabolomic Analyses to Map Phenolic Acid Synthesis Pathways

We jointly analyzed metabolomics and transcriptomics to map the pathway of phenolic acid biosynthesis. Orange bars show the accumulation of primary metabolites in the pathway map, which are L-phenylalanine, L-tyrosine and phenylpyruvic acid (Figure 4a). Purple bars show the accumulation of differential phenolic acids (Figure 4b), and the expression of related genes is shown as a heatmap in the pathway map (Figure 4c). The pathway map can show the upstream and downstream relationship of each metabolite and the expression of related genes, which helps to study the relationship between abscisic acid and phenolic acid synthesis.

### 2.6. Significant Changes in Phenolic Acids in Externally Applied Abscisic Acid in R. chrysanthum

To further investigate the relationship between abscisic acid and phenolic acid synthesis, the present study investigated the changes in phenolic acids under UV-B stress in externally applied abscisic acid in *R. chrysanthum*. Principal component analysis showed that principal component 1 explained 28.44% of the data set and principal component 2 explained 16.8% of the data set, and the NQ groups were able to separate well (Appendix A). There were 289 differential metabolites between the NQ groups, with 106 metabolites increased and 183 metabolites decreased (Appendix A). The categorization and number of differential metabolites are shown in Figure 5a. Fifty-one phenolic acids produced significant changes after the external application of ABA, which may promote the response of *R. chrysanthum* to UV-B stress by affecting the accumulation of phenolic acids. We constructed a heatmap of these 51 differential phenolic acids, showing differences in phenolic acid content between the NQ groups (Figure 5b). The differential phenolic acid information for the top-20 differential multiplicity is shown in Figure 5c. 5-methoxysalicylic acid, 2-phenylethy-1-o-β-d-glucoside and 1-o-(3,4-dihydroxy-5-methoxy-benzoyl)-glucoside were significantly decreased. Glucovanillin, 1,3,6-tri-o-galloyl-β-d-glucose and 1-o-p-coumaroyl-β-d-glucose were significantly increased and may be the key metabolites in response to UV-B stress under ABA treatment. The box-and-line plot demonstrates the amount of ABA detected in the NQ group (Figure 5d). Correlation analysis between ABA and differential phenolic acids showed that ABA was negatively correlated with methyl syringate and 1-o-caffeoyl-β-d-glucose* and positively correlated with all others (Figure 5e) (Appendix A). The correlation of ABA with differential phenolic acids that could be annotated for the KEGG database is shown in Figure 5f (Appendix A). Information on phenolic acids that significantly positively responded to UV-B stress after ABA treatment is shown in Table 2.

### 2.7. K-Means Analysis of the Three Treatment Groups Revealed That Phenolic Acids Were Affected by ABA

K-means analysis of phenolic acids from samples in groups M, N and Q revealed that the 394 phenolic acids could be divided into seven clusters according to the trend of accumulation. The highest number of Cluster 1 was 251 and the lowest number of Cluster 6 was 1, suggesting that there were differences in phenolic acid accumulation among treatments, which might be related to the plants’ response to the changing environment (Figure 6a). We counted the number of metabolites that differed between the comparison groups in each of the seven clusters, and the results are shown in Figure 6b. Because abscisic acid promotes plant response to stressful environments, phenolic acids that differed significantly in MN and not in MQ may be metabolites responsive to UV-B stress under abscisic acid treatment, and we show the content of these metabolites in a box-and-line plot (Figure 6c). Acteoside, 1,3,6-tri-o-galloyl-β-d-glucose, picein and 1-o-caffeoyl-β-d-glucose* trends belong to Cluster 1, glucovanillin trends belong to Cluster 2 and 1-o-feruloyl-β-d-glucose, syringin, salidroside, arbutin and 1,6-di-o-galloyl-β-d-glucose trends belong to Cluster 4. Information about these 10 metabolites is shown in Table 3.

### 2.8. Hypothetical Modeling of Phenolic Acid Response of R. chrysanthum Leaves under UV-B Stress

We constructed a hypothetical model of the phenolic acid response in leaves of *R. chrysanthum* under UV-B stress. The left side indicates the phenolic acid changes in response to UV-B radiation in the MN group, and the right side indicates the phenolic acid changes in response to UV-B radiation in *R. chrysanthum* applied with ABA. It was found that externally applied abscisic acid in *R. chrysanthum* was able to increase tolerance to UV-B radiation by affecting phenolic acid production. This model helps to increase our understanding of plant resistance mechanisms (Figure 7).

## 3. Discussion

Plant resistance or tolerance to environmental stresses can affect the productivity of ecosystems, and the destruction of the ozone layer has led to a massive increase in the amount of ultraviolet light reaching the Earth’s surface, increasing the stress on plants [27]. Plants have evolved resilience mechanisms suitable for survival during environmental stress, such as biochemical reprogramming and fixed morphological structures [28]. The ability of the epidermal wax layer present on the leaves to reflect high-intensity UV radiation is one of its manifestations [29]. Using a combination of metabolomics and proteomics, one study found that UV-B radiation promotes the conversion of primary metabolites to phenolics in the leaves of the *R. chrysanthum*, which may be responsible for its resistance to UV-B radiation [30]. We, therefore, investigated what adaptive changes occurred in the leaves of *R. chrysanthum* under stress conditions.

In this study, we performed transcriptomic and metabolomic analyses of *R. chrysanthum* leaves cultured in an artificial climate chamber simulating an alpine environment. We identified 522 differential metabolites and 2348 DEGs in *R. chrysanthum* that received and did not receive UV-B radiation (groups M and N). Differential metabolites are mainly concentrated in “ABC transporters”, “amino acid biosynthesis”, “D-amino acid metabolism” and “phenylalanine metabolism”. ABC transporter proteins are capable of the transmembrane transport of a wide range of substances, can participate in signal transduction in eukaryotes and can regulate cell membrane permeability [31]. The enrichment of differential metabolites in the ABC transporter pathway after UV-B stress may be an important manifestation of the transmembrane transport of secondary metabolites produced by *R. chrysanthum* and, thus, adapt to the stress. Phenylalanine is a key node in primary and secondary metabolism and is a precursor for the synthesis of many secondary metabolites, which play important roles in plant response to environmental stresses [32]. Among them, there were 115 differential phenolic acids and 672 genes related to phenolic acids, suggesting that phenolic acids may be the key substances in the response of *R. chrysanthum* leaves to UV-B stress. Abscisic acid accumulation was significantly altered after UV-B stress, and abscisic acid plays an important role as a signaling molecule in regulating plant growth and development. We performed metabolomics assays on externally applied ABA and non-ABA-applied *R. chrysanthum* (groups N and Q) to verify the important role of ABA in the plant’s resistance to UV-B stress. We found that ABA was able to influence phenolic acid accumulation under UV-B radiation and enhance tolerance to stress. These results suggest that UV-B stress induced an integrated defense response in the leaves of *R. chrysanthum*, including the enrichment of antioxidant compounds and phytohormone signaling.

### 3.1. Phenolic Acid Enhances UV-B Radiation Resistance in R. chrysanthum

*A. sibirica* is considered to have a strong antioxidant capacity due to its high levels of phenolic compounds [33]. Salvianolic acid B is able to act as an antioxidant by providing hydrogen atoms to scavenge oxygen free radicals and by regulating the expression of antioxidant enzymes to reduce the production of oxygen free radicals and oxygen-containing non-radicals [34]. Phenolic acids were significantly altered in the leaves of *R. chrysanthum* after UV-B radiation, and their metabolomic data demonstrated the accumulation of differential phenolic acids. Leucosceptoside A, 4-o-digalloyl-3,5-di-o-galloylquinic acid and 3-o-feruloylquinic acid-o-glucoside are significantly increased phenolic acids after UV-B radiation. 2,4,6-tri-o-galloyl-d-glucose, 2,3-di-o-galloyl-β-d-glucose* and raspberryketone glucoside are significantly decreased phenolic acids. Leucosceptoside A is a phenylethanol glycoside with hypoglycemic and antihypertensive activities. Leucosceptoside A has inhibitory activity against alpha-glucosidase and PKCα [35]. Leucosceptoside A has been less studied in plants, but its significant accumulation under stress conditions suggests resistance to UV-B radiation. 4-o-digalloyl-3,5-di-o-galloylquinic acid and 3-o-feruloylquinic acid-o-glucoside also play important functions against UV-B stress. Raspberryketone glucoside is able to be synthesized from isolindleyin, a tyrosinase inhibitor with anti-inflammatory, analgesic and anti-melanogenic activities [36]. Among the phenolic acids that underwent significant differences after UV-B stress, seven phenolic acids were significantly correlated, and 2-hydroxy-3-phenylpropanoic acid, anthranilic acid, 2-hydroxycinnamic acid*, 3-hydroxyphenylacetic acid, 2-phenylethanol, 3-hydroxycinnamic acid* and α-hydroxycinnamic acid* were able to demonstrate this in a metabolic pathway. All seven phenolic acids were significantly increased, significantly corresponding to UV-B radiation.

Our transcriptomic data showed that 672 genes could annotate phenol acid-related pathways after UV-B radiation, and these genes could exert binding and catalytic activities. In the cellular amino acid metabolic process, oxidoreductase activity, L-aspartate:2-oxoglutarate aminotransferase activity, pyridoxal phosphate binding and NAD binding are significantly enriched, which are related to the synthesis of secondary metabolites and play an important role in phenylalanine metabolism. We analyzed and screened these genes for accumulation patterns and obtained eight genes that were significantly altered in the phenylalanine metabolic pathway. TRINITY_DN104_c0_g1_i10-C_1 and TRINITY_DN104_c0_g1_i2-C_1 were significantly downregulated after UV-B radiation, and both genes were able to encode L-tryptophan decarboxylase together with TRINITY_DN30625_c0_g1_i1-C_3, which catalyzes the conversion of L-phenylalanine to name phenethylamine, which, in turn, leads to the formation of significantly altered 3-hydroxyphenylacetate and phenylethyl alcohol through a multistep reaction, and these significantly altered phenolic acids are able to respond to UV-B stress. The regulatory network of genes and metabolites associated with phenolic acids was mapped, demonstrating adaptive responses to UV-B stress.

### 3.2. Abscisic Acid Plays an Important Role in UV-B Radiation Resistance in R. chrysanthum

ABA is a hemipterpenoid phytohormone that regulates developmental processes such as water conductance and stomatal closure and plays a central role in adaptive responses to abiotic stresses, such as low temperature and drought [37]. Transcriptomics and proteomics studies of *R. chrysanthum* after UV-B stress revealed that ABA signaling in the plant was able to partially counteract the damage caused by UV-B radiation by regulating stomatal closure through associated phosphorylated proteins [38]. Endogenous ABA accumulation is regulated by a balance between biosynthesis and catabolism [39]. Under drought stress, ABA content in alginate-treated tomatoes was lower than that of the control, and alginate selectively inhibited ABA biosynthesis and promoted ABA catabolism, suggesting that alginate is involved in the negative regulation of ABA metabolism under drought stress [40]. Previous studies have also shown that exogenous alginate treatment reduces endogenous ABA content in soybean under drought conditions [41]. Aldehyde 3-phosphate can produce 1-deoxy-d-xylulose 5-phosphate, which can then produce ABA through the MEP/DOXP pathway. Glyceraldehyde 3-phosphate can also produce L-phenylalanine through a multistep reaction. Thus, the decrease in ABA content after UV-B radiation may be associated with an increase in L-phenylalanine content, and the accumulation of L-phenylalanine affects phenolic acid content, which is consistent with the fact that ABA acts through signaling rather than content under adversity stress. ABA signaling is a double-negative regulatory mechanism, in which ABA binds to PYR/PYL, thereby inhibiting PP2C activity, and PP2C inactivates SnRK2 [42]. In the presence of ABA, ABA binds to the receptor and then interacts with PP2C, allowing for SnRK2 activation and target protein phosphorylation, consistent with previous studies [38]. ABA with differential phenolic acids showed positive correlation with 14 phenolic acids (where ABA was significantly positively correlated with syringin and picein) and negative correlation with 13 phenolic acids. Syringin is able to exert antioxidant effects by regulating the SIRT1 signaling pathway, and there are fewer studies on syringin in plants, but the increase in syringin, which was significantly and positively correlated with ABA in the present study, was an adaptive response of *R. chrysanthum* to UV-B stress. The relationship between ABA and phenolic acid synthesis is shown in the pathway heatmap, demonstrating the ability of ABA to influence the response of *R. chrysanthum* to UV-B stress by affecting phenolic acid synthesis.

### 3.3. External Application of Abscisic Acid Gives Greater Resistance to Radiation in R. chrysanthum

In order to investigate the role of ABA in the defense of *R. chrysanthum* against UV-B radiation, metabolomics studies revealed that externally applied ABA produced more phenolic acids (51) that varied differentially between *R. chrysanthum* and untreated *R. chrysanthum* that received the same dose of UV-B radiation at the same time. 2-phenylethanol, 1-o-p-coumaroyl-β-d-glucose, 2-hydroxy-3-phenylpropanoic acid, 3-(4-hydroxyphenyl)-propionic acid and 3-o-feruloylquinic acid-o-glucoside were significantly increased in the MN and NQ groups, indicating that these five phenolic acids were the compounds that significantly responded to UV-B stress in the leaves of *R. chrysanthum* and were able to increase the stress tolerance of *R. chrysanthum* and positively regulated by ABA. The phenolic content of dandelion petals has antioxidant activity and is beneficial for oxidative stress [43]. Studies on hawthorn fruits revealed that 32 phenolic compounds were significantly upregulated in WLZR, which may be related to the higher antioxidant activity in WLZR [44]. Plants utilize different modes of ABA signaling strength determinants to cope with different stress conditions, and, thus, the response to stress conditions may also vary among plants [45]. Studies on papaya have shown that *MpAITR1* positively regulates the ABA signaling pathway and enhances tolerance to drought stress by modulating downstream target genes [46]. Studies on exogenously applied ABA in wheat, potato and winter oilseed rape suggest that ABA and cold may act in a common physiological process, thereby enhancing frost resistance in plants [47]. Similarly, ABA and UV-B radiation may act in a common physiological process, thereby enhancing radiation tolerance in externally applied ABA in *R. chrysanthum*.

### 3.4. Abscisic Acid Can Influence Phenolic Acid Production to Promote the Resistance of R. chrysanthum to UV-B Stress

In order to find the metabolites that responded significantly to UV-B stress, metabolite assays were performed on *R. chrysanthum* in groups M, N and Q. K-means results showed seven trends in metabolite accumulation among the three groups. The MN intergroup results indicated that the *R. chrysanthum* was able to respond to UV-B stress by altering the content of phenolic acids. The results of the NQ group showed that ABA was able to influence phenolic acid production. Phenolic acids that differed significantly between the MN groups and not between the MQ groups may be ABA-influenced phenolic acids. Thus, the *R. chrysanthum* in group Q showed greater tolerance to UV-B radiation. Under UV-B radiation, there were differences in the effects of ABA on different phenolic acids, with positive regulation of 1,3,6-tri-o-galloyl-β-d-glucose, glucovanillin and 1,6-di-o-galloyl-β-d-glucose, and negative regulation of picein and 1-o-Caffeoyl-β-d-glucose*. ABA-pretreated rice seedlings were studied, and ABA was found to reduce leaf damage to abiotic stress at the proteome level [48]. 24-epibrassinolide (EBR) was able to positively regulate drought response in plants, and it was found that EBR promoted ABA accumulation and improved drought tolerance in tea trees [49]. Similarly, in this study, phenolic acids were found to be able to respond to UV-B stress, and externally applied ABA in *R. chrysanthum* was able to increase its tolerance to UV-B radiation by affecting phenolic acid accumulation. In the present study, we concluded that abscisic acid can influence phenolic acid production to improve tolerance to UV-B stress in *R. chrysanthum.*

## 4. Materials and Methods

### 4.1. Plant Materials and Treatments

*Rhododendron chrysanthum* Pall. was collected from Changbai Mountain at an altitude of 1300–2650 m and preserved in artificial climate chambers (Ningbo Saifu Experimental Instrument Co., Ningbo, China) that simulate the alpine environment. The climate chamber environment was 16–18 °C, 14 h of light and 10 h of darkness and 60% relative humidity, and white-fluorescent lamps were used at 50 µmol (photons) m^−2^ s^−1^.

Uniformly grown 8-month-old seedlings were selected and divided into three groups, M, N and Q, with three biological replicates each. Groups M and N were transplanted into 1/4 MS medium, and group Q was transplanted into 1/4 MS medium externally applied with ABA (100 µmol/L). After one week of incubation, radiation treatment was carried out for a period of 2 days (8 h per day).

Group M was treated with PAR (400–700 nm light required by plants for photosynthesis), and groups N and Q were treated with UV-B (280–315 nm). Filters (Edmund, Filter Long 2IN SQ, Barrington, NJ, USA) with different transmittances were placed over the vials for radiation treatment, with 400 nm filters used for PAR treatment and 295 nm filters for UV-B treatment. The outside of the culture flask was wrapped in tin foil and labeled. PAR is provided by warm white-fluorescent lamps (Philips, T5 × 14 W, Amsterdam, The Netherlands) and UV-B by UV-B fluorescent tubes (Philips, Ultraviolet-B TL 20 W/01 RS, Amsterdam, The Netherlands). The radiation process lasted for 2 days (8 h per day). The irradiance of the samples that effectively received UV-B treatment was 2.3 Wm^−2^ UV-B, according to the transmission function of the long-pass filter, which was measured with a UV intensity meter (Sentry Optron-ICS Corp., ST-513, SHH, New Taipei City, China) and a light meter (TES Electrical Electronic Corp., Tes-1339 Light Meter Pro., Taipei, China) [10,50].

After UV-B treatment for 2 days, leaf samples were collected in liquid nitrogen for RNA-seq analysis and widely targeted metabolomics assays. Material handling methods are shown in Appendix A.

### 4.2. Widely Targeted Metabolomics Assays

In this experiment, metabolomics was examined on leaves of groups M, N and Q. The leaves preserved in liquid nitrogen were lyophilized and processed using a grinder (MM 400, Retsch), electronic balance (MS105DΜ), vortexer and centrifuge, and the samples filtered through a microporous filter membrane (0.22 μm pore size) were preserved in injection bottles for UPLC-MS/MS analysis.

The column, mobile phase, elution gradient, flow rate, etc., in the liquid-phase conditions are set according to the company’s requirements. The mass spectrometry conditions mainly include the following: electrospray ionization (ESI) temperature 500 °C; ion spray voltage (IS) 5500 V (positive ion mode)/−4500 V (negative ion mode); ion source gas I (GSI), gas II (GSII) and curtain gas (CUR) were set to 50, 60, and 25 psi, respectively; and collision-induced ionization parameters were set to high. QQQ scans were performed using MRM mode, with the collision gas (nitrogen) set to medium. DP and CE of individual MRM ion pairs were accomplished by further de-clustering potential (DP) and collision energy (CE) optimization. A specific set of MRM ion pairs was monitored in each period based on the metabolites eluted during each period.

Metabolite quantification was accomplished using triple-quadrupole mass spectrometry in multiple reaction monitoring (MRM) mode of analysis. After obtaining the metabolite profiling data of different samples, peak area integration was performed for all the substance chromatographic peaks, and the integration was corrected for the mass spectral peaks of the same metabolite in different samples among them [51]. Mass spectrometry data were processed using the software Analyst 1.6.3.

### 4.3. RNA-Seq Analysis

Transcriptome sequencing was performed using the sequencing platform of Shenzhen Huada Gene Science and Technology Research Co. (Shenzhen, China). Raw data from sequencing contain low quality, splice contamination and reads with too much unknown base N. These reads need to be removed prior to data analysis to ensure the reliability of the results. This project uses UW’s self-developed filtering software SOAPnuke (v1.6.5) for filtering. Total Clean Reads were in the range of 42–44 M, and as a whole, the proportion of low-quality (quality < 20) bases was low, indicating good sequencing quality. Clean reads were compared to reference gene sequences using Bowtie2 (v2.2.5), followed by RSEM to calculate gene and transcript expression levels [52]. Raw transcriptome data were deposited in the Sequence Read Archive with the accession number PRJNA756577 (https://www.ncbi.nlm.nih.gov/sra/PRJNA756577, accessed on 1 September 2023).

### 4.4. De Novo Assemble

We used Trinity (v2.0.6) to de novo assemble clean reads (removing PCR repeats to improve assembly efficiency), and then used CD-HIT (4.6) to cluster and de-redundantly cluster the assembled transcripts to obtain Unigene. Trinity consists of three separate modules: Inchworm, Chrysalis, and Butterfly. These process large numbers of reads in turn. Trinity first constructs the reads into a large number of individual de Bruijn plots and then extracts full-length transcript shear isoforms for each plot separately. The quality of the assembled transcripts was assessed using the single-copy direct homology database BUSCO (https://busco.ezlab.org/, accessed on 15 August 2023), and the results of the assembly evaluation were improved by comparing with conserved genes, which, to a certain extent, indicates the completeness of the transcriptome assembly.

### 4.5. Gene Annotation

The assembled Unigene will be annotated in seven functional databases (KEGG, GO, NR, NT, SwissProt, Pfam and KOG), with 41,815 entries in the KEGG database, 38,760 entries in the GO database, 66,130 entries in the NR database, 45,228 entries in the NT database, 41,317 entries in the SwissProt database, 40,557 entries in the Pfam database and 33,202 entries in the KOG database. The KEGG database is able to visualize the metabolic pathways of interest by using graphics to present the numerous metabolic pathways and the relationships between the pathways. The KEGG Orthology (KO) system provides a cross-species annotation process by linking relevant information from molecular networks into the genome. Comparing genes with different databases enables us to understand the structure and function of genes in different ways.

### 4.6. Screening for Differential Metabolites and Differentially Expressed Genes

Screening for differential metabolites was based on the following two criteria: (1) metabolites with VIP (variable importance in projection) > 1 were selected; the VIP value indicates the strength of the effect of between-group differences for the corresponding metabolite on the categorical discrimination of the samples in each group in the model, and metabolites with VIP > 1 were generally considered to be significantly different; (2) metabolites with FC (fold change) ≥ 1.5 and FC ≤ 0.67 were selected. Metabolites with a FC ≥ 1.5 or ≤0.67 and VIP > 1 in the control and experimental groups were considered significant.

We used Bowtie2 (v2.2.5) to compare clean reads to genomic sequences and then used RSEM (v1.2.8) to calculate gene expression levels for individual samples [53]. The DEseq2 method is based on the principle of negative binomial distribution, and this project performs differentially expressed gene detection according to the method described in Michael I et al. [54]. The parameter used was Qvalue (Adjusted *p* value) < 0.05. In this study, the screening criteria for differentially expressed genes were set as FC > 1 and Qvalue < 0.05.

### 4.7. Statistical Analysis for Bioinformatics

Principal component analysis (v2.0) provides an intuitive interpretation of complex data sets, revealing groupings and trends in the observations in the data set and can also be used to tease out outlier samples. Boxplots were drawn using the analysis package Python 3.6.6 and the function package pandas 0.23.4. Combined correlation plots were plotted using the ggcor package (0.9.8.1) for correlation plots and mantal analysis. Kmeans_cluster (v2.2) is able to perform clustering based on expression and sample grouping information and visualize the clustering results, and the same cluster set has the same expression pattern in multiple groupings. The analysis package used for Kegg enrichment analysis was R version 3.5.1, and the function package was ggplot2 3.3.0.

## 5. Conclusions

In this study, we used *R. chrysanthum* in a simulated alpine environment to investigate the adaptive responses of alpine plants to UV-B stress. Using widely targeted metabolomics and transcriptomics, we investigated the key metabolic processes in response to UV-B stress in *R. chrysanthum*, constructed a pathway map of phenolic acid synthesis and studied the effects of externally applied ABA on UV-B stress in *R. chrysanthum*. The ability of phytohormones to play a role in plant stress resistance has been confirmed by many researchers. And this paper, using a unique species and the latest research methodology, can add information on this topic for the species of *R. chrysanthum*. It was found that UV-B stress led to changes in phenolic acids and related genes in response to UV-B stress. The externally applied ABA in *R. chrysanthum* had high tolerance to UV-B radiation, and its phenolic acid content in the face of the stressful environment was significantly different from that of *R. chrysanthum* with no ABA applied, indicating that the ABA pretreatment could improve its radiation tolerance. In this study, we investigated the response of *R. chrysanthum* to UV-B stress from the perspectives of hormones and secondary metabolites, and we found that abscisic acid can influence phenolic acid production to improve tolerance to UV-B stress in *R. chrysanthum*, but the complex molecular regulatory mechanisms still need to be further investigated.

## Figures and Tables

**Figure 1 ijms-25-01234-f001:**
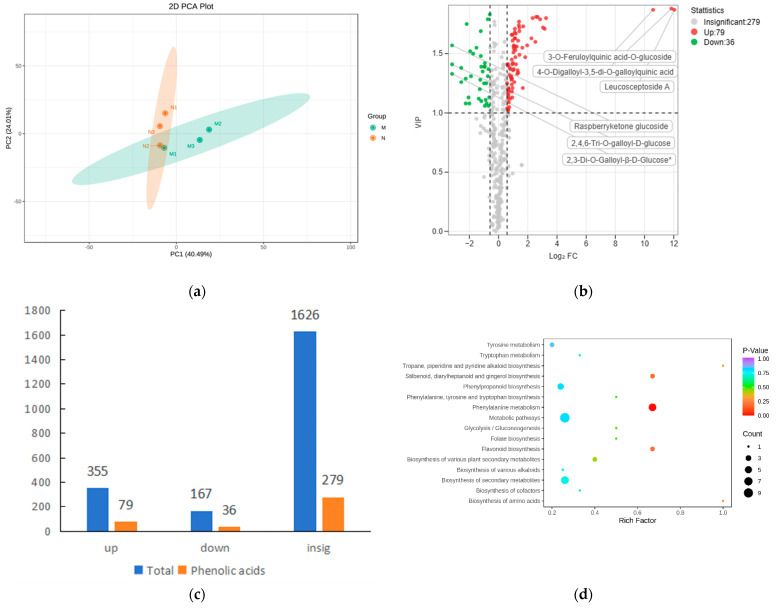
Significant changes in phenolic acid content of *R. chrysanthum* leaves after UV-B stress. (**a**) Phenolic acid PCA chart; (**b**) MN group volcano map screening for differential phenolic acids; (**c**) total metabolite and phenolic acid quantity Statistics; (**d**) bubble diagram for KEGG enrichment analysis of MN group phenolic acids; (**e**) MN group differential phenolic acid heatmap; (**f**) differential phenolic acid thermograms between MN groups able to annotate to the KEGG database; (**g**) MN group differential phenolic acid infographic. The red color in the volcano plot indicates elevated metabolite levels, green color indicates decreased metabolite levels, and the horizontal dashed line indicates a VIP value of 1, while the vertical dashed line indicates FC ≥ 1.5 or ≤0.67. The more purple colors in the heatmap indicate higher levels of metabolites, and the more orange colors indicate lower levels of metabolites. In the lollipop chart, the size of the dots indicates the VIP value, and the horizontal coordinate is the multiplicity of differences, with metabolite increase in red and metabolite decrease in green. The “*” in the lollipop plot indicates *p* < 0.05 for metabolites, “**” indicates *p* < 0.01 for metabolites, and the “*” after metabolites is just the method of displaying metabolites and is not statistically significant.

**Figure 2 ijms-25-01234-f002:**
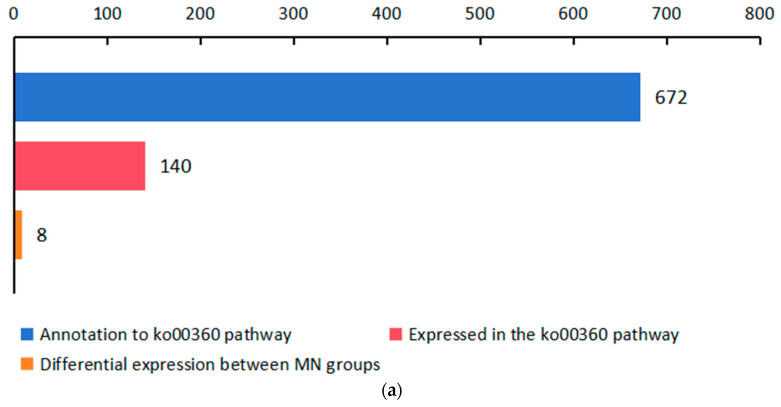
Significant changes in genes related to phenolic acid synthesis in leaves of the *R. chrysanthum* after UV-B stress. (**a**) Number of genes statistics; (**b**–**e**) 672 genes GO, KEGG classification and enrichment; (**f**) heatmap of genes expressed in the ko00360 pathway; (**g**) heatmap of DEGs; (**h**) expression of DEGs. Red circles in (**g**) indicate Q-value < 0.05 between MN groups. Small letters a and b indicate significant differences (*p* < 0.05). Values are means ± S.E. (n = 3). DDC: L-tryptophan decarboxylase; AOC3: primary-amine oxidase; E3.5.1.4: amidase; PAAS: phenylacetaldehyde synthase; TAT: tyrosine aminotransferase; HPD: 4-hydroxyphenylpyruvate dioxygenase.

**Figure 3 ijms-25-01234-f003:**
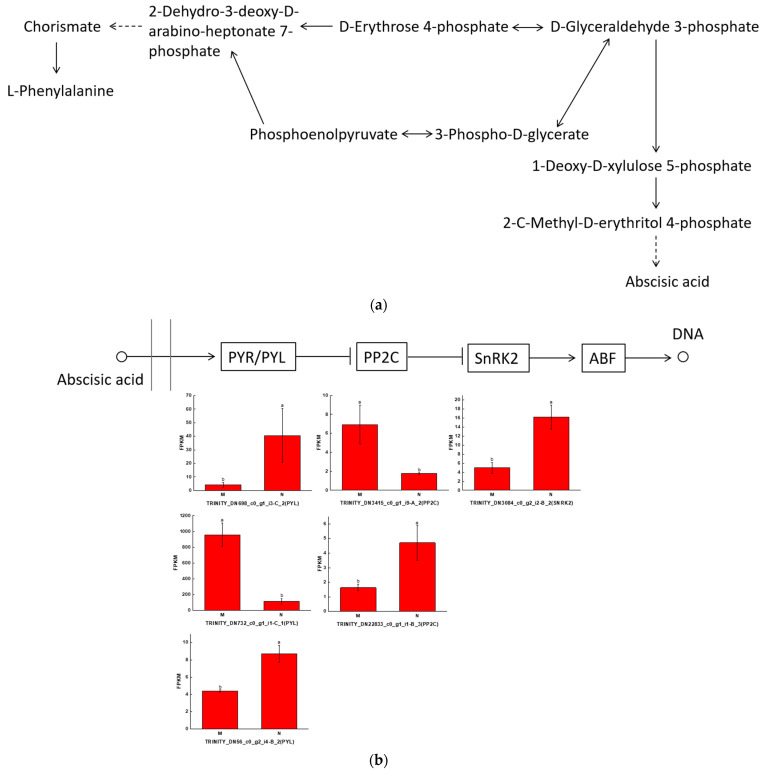
Strong correlation between phenolic acid and abscisic acid in leaves of the *R. chrysanthum*. (**a**) Relationship of L-phenylalanine to abscisic acid; (**b**) abscisic acid signal transduction; (**c**) correlation analysis of abscisic acid with differential phenolic acid. Small letters a and b indicate significant differences (*p* < 0.05). The color of the squares indicates the Pearson correlation coefficient, with redder colors indicating stronger positive correlations and bluer colors indicating stronger negative correlations. The lines on the left side represent the Mantel’s statistic, the thickness of the lines reflects the correlation between the hormones and the corresponding differential metabolites, and the color of the lines represents the degree of significance. “*” is just a presentation of metabolites identified using widely targeted metabolomics with no statistical significance.

**Figure 4 ijms-25-01234-f004:**
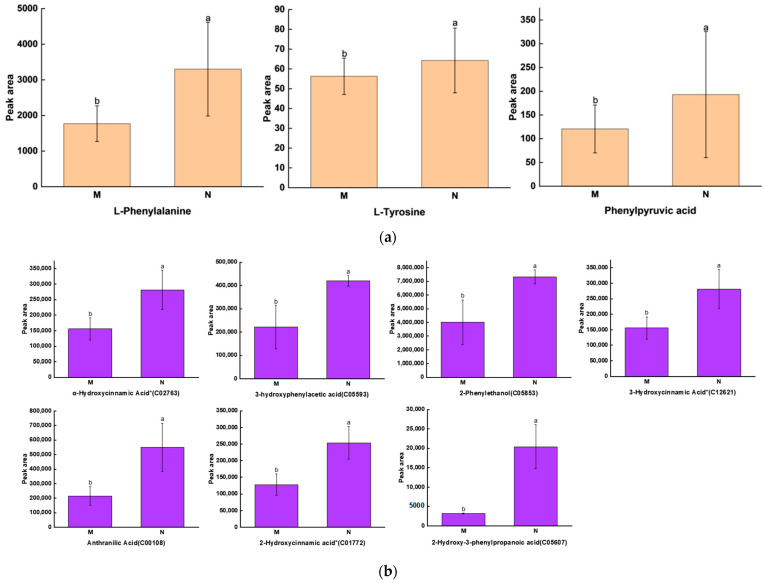
Phenolic acid synthesis pathway under UV-B stress. (**a**) Expression of precursors for phenolic acid synthesis; (**b**) expression of differential phenolic acids; (**c**) heatmap metabolic pathway modeling of DEG expression patterns of the *R. chrysanthum* under UV-B radiation. Small letters a and b indicate significant differences (*p* < 0.05). A more orange color indicates a higher metabolite level, and a bluer color indicates a lower metabolite level. Redder colors indicate higher gene expression and greener colors indicate lower gene expression. “*” is just a presentation of metabolites identified using widely targeted metabolomics with no statistical significance.

**Figure 5 ijms-25-01234-f005:**
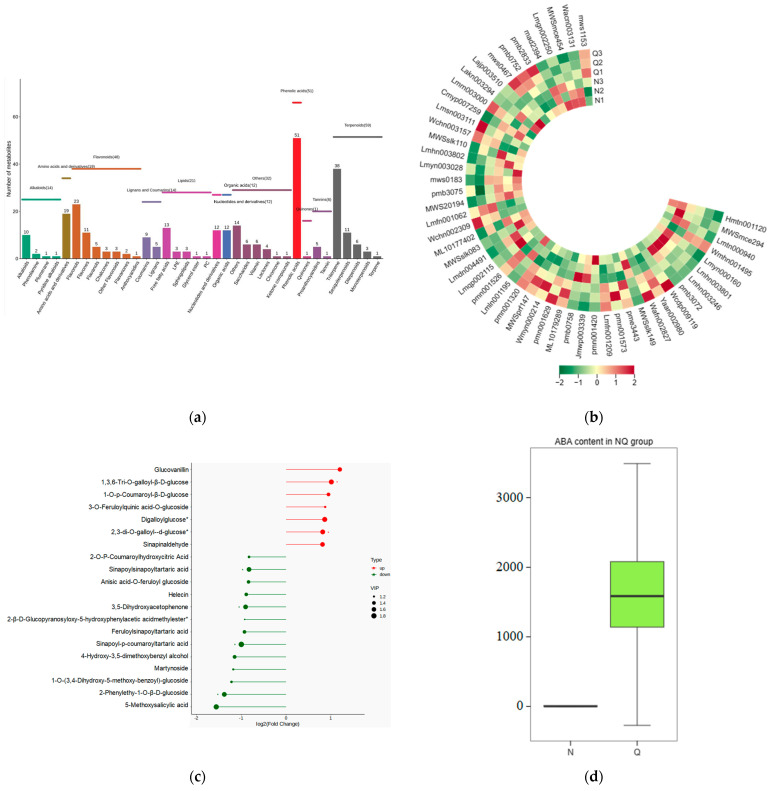
Significant changes in phenolic acids in leaves of *R. chrysanthum* after external application of ABA. (**a**) Classification and quantity statistics of differential metabolites in the NQ group; (**b**) heatmap of differential phenolic acid expression in the NQ group; (**c**) information on the expression of the first 20 differential phenolic acids in the NQ group; (**d**) boxplot of ABA expression for NQ group; (**e**) correlation analysis of abscisic acid with 51 differential phenolic acid; (**f**) correlation analysis of abscisic acid with 15 differential phenolic acid. In the heatmap, redder colors indicate higher metabolite levels and greener colors indicate lower metabolite levels. The color of the squares indicates the Pearson correlation coefficient, with redder colors indicating stronger positive correlations and bluer colors indicating stronger negative correlations. The lines on the left side represent the Mantel’s statistic, the thickness of the lines reflects the correlation between the hormones and the corresponding differential metabolites, and the color of the lines represents the degree of significance. The “*” in the lollipop plot indicates *p* < 0.05 for metabolites, and the “*” after metabolites is just the method of displaying metabolites and is not statistically significant.

**Figure 6 ijms-25-01234-f006:**
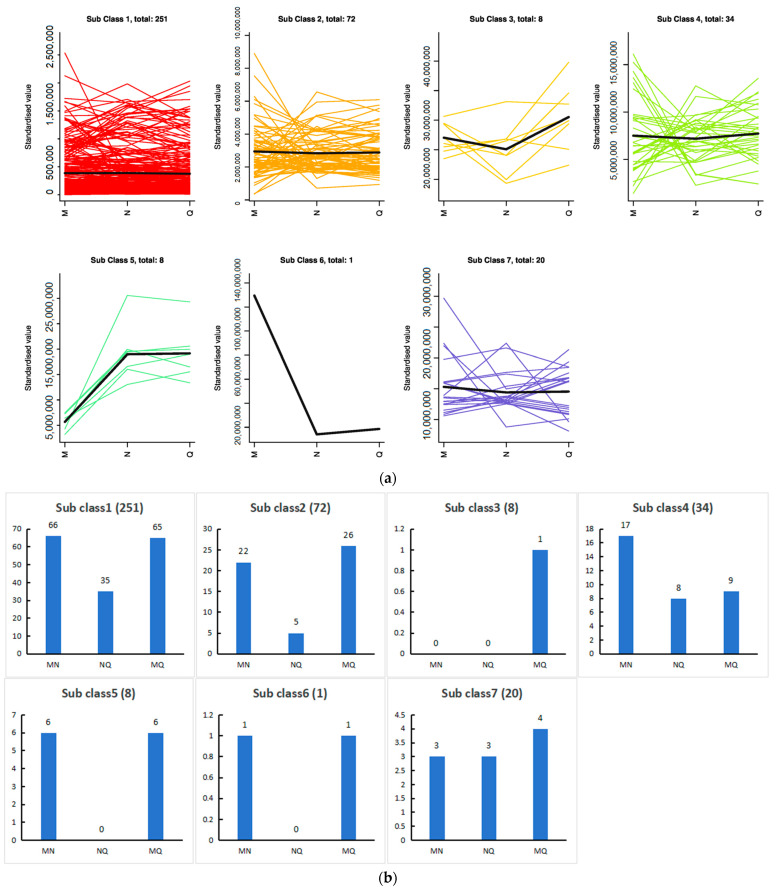
Metabolomics analysis of the MNQ group. (**a**) K-means analysis; (**b**) statistics on the number of differential metabolites corresponding to clusters in K-means analysis; (**c**) boxplot of potentially eligible metabolite expression. “*” is just a presentation of metabolites identified using widely targeted metabolomics with no statistical significance.

**Figure 7 ijms-25-01234-f007:**
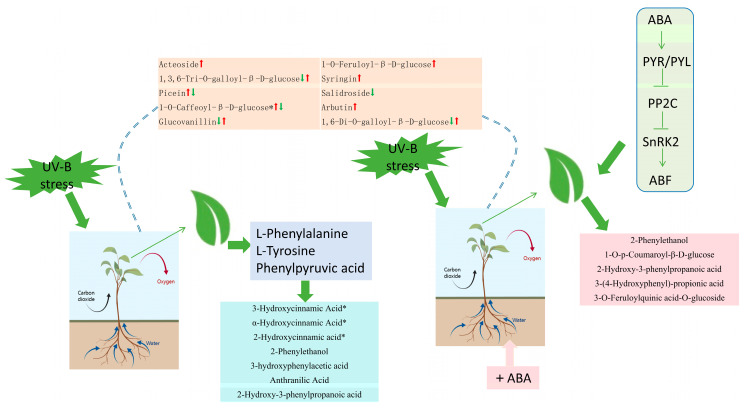
Modeling hypotheses for phenolic acid biosynthesis under UV-B stress in leaves of the *R. chrysanthum*. The first arrow indicates the differential expression situation between the MN groups and the second arrow indicates the differential expression situation between the NQ groups. “*” is just a presentation of metabolites identified using widely targeted metabolomics with no statistical significance.

**Table 1 ijms-25-01234-t001:** Widely targeted metabolomics assay obtained 2148 metabolites.

Metabolite Categories	Class I	Class II	Number of Metabolites
Primary metabolites (505)	Amino acids and derivatives (188)	Amino acids and derivatives	188
	Nucleotides and derivatives (75)	Nucleotides and derivatives	75
	Organic acids (108)	Organic acids	108
	Lipids (134)	Glycerol ester	14
		PC	2
		Sphingolipids	8
		LPC	12
		LPE	24
		Free fatty acids	74
Secondary metabolites (1372)	Alkaloids (117)	Pyridine alkaloids	6
		Phenolamine	17
		Quinoline alkaloids	7
		Alkaloids	61
		Isoquinoline alkaloids	5
		Plumerane	18
		Piperidine alkaloids	3
	Flavonoids (487)	Anthocyanidins	5
		Chalcones	15
		Flavanols	38
		Flavanones	46
		Flavanonols	10
		Flavones	200
		Flavonols	139
		Isoflavones	12
		Other Flavonoids	22
	Phenolic acids (394)	Phenolic acids	394
	Quinones (17)	Anthraquinone	10
		Quinones	7
	Lignans and Coumarins (91)	Lignans	49
		Coumarins	42
	Tannins (37)	Tannin	20
		Proanthocyanidins	17
	Terpenoids (227)	Sesquiterpenoids	58
		Monoterpenoids	44
		Ditepenoids	34
		Triterpene	88
		Terpene	3
	Steroids (2)	Steroid	2
Others (271)	Others (271)	Alcohol compounds	10
		Lactones	10
		Others	96
		Aldehyde compounds	14
		Chromone	11
		Saccharides	79
		Ketone compounds	17
		Vitamin	29
		Stilbene	5

**Table 2 ijms-25-01234-t002:** Phenolic acids significantly and positively responding to UV-B stress after ABA treatment.

Compounds	Structure	MN Fold_Change	NQ Fold_Change
2-phenylethanol	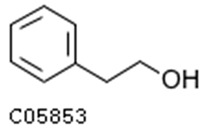	1.82	1.65
1-o-p-coumaroyl-β-d-glucose	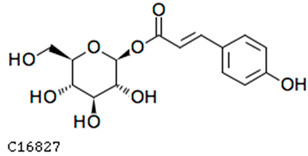	1.86	1.93
2-hydroxy-3-phenylpropanoic acid	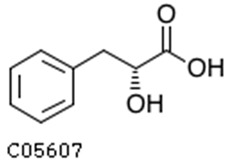	6.39	1.61
3-(4-hydroxyphenyl)-propionic acid	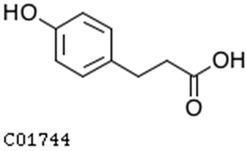	2.07	1.59
3-o-feruloylquinic acid-o-glucoside	-	1538.86	1.83

**Table 3 ijms-25-01234-t003:** Information about key metabolites.

Compounds	cpd_ID	Formula	Exact Mass	Mol Weight	Structure
acteoside	C10501	C29H36O15	624.2054	624.5871	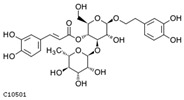
1,3,6-tri-o-galloyl-β-d-glucose	C17458	C27H24O18	636.0963	636.4687	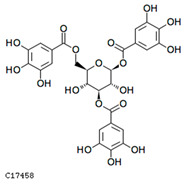
picein	C10720	C14H18O7	298.1053	298.2885	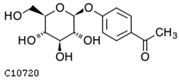
1-o-caffeoyl-β-d-glucose*	C10433	C15H18O9	342.0951	342.298	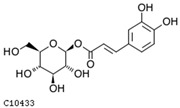
glucovanillin	C19808	C14H18O8	314.1002	314.2879	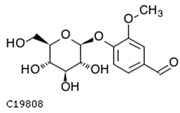
1-o-feruloyl-β-d-glucose	C17759	C16H20O9	356.1107	356.3246	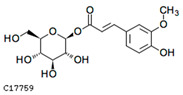
syringin	C01533	C17H24O9	372.142	372.3671	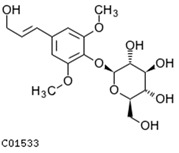
salidroside	C06046	C14H20O7	300.1209	300.3044	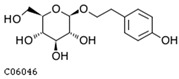
arbutin	C06186	C12H16O7	272.0896	272.2512	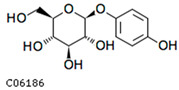
1,6-di-o-galloyl-β-d-glucose	C04101	C20H20O14	484.0853	484.3644	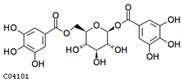

Note: “*” is just a presentation of metabolites identified using widely targeted metabolomics with no statistical significance.

## Data Availability

The data used in this study are available from the corresponding author on submission of a reasonable request.

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
