# Peer review of "Abscisic Acid Affects Phenolic Acid Content to Increase Tolerance to UV-B Stress in Rhododendron chrysanthum Pall."

_ijms, 2024, doi:10.3390/ijms25021234_

Round 1
Reviewer 1 Report
Comments and Suggestions for Authors
The available test of manuscript is well presented,
the mathematical part of the study presented in detail. The results presentation is clear and concise.
Disadvantages:
- In the “Introduction”, it is necessary to emphasize more clearly the relevance of the current study and its practical significance
- It is necessary to decipher all abbreviations set out in the text of the article and tables (Table 2) at the first mention.
- In Figure 1 А, it is unclear which of the studied groups is represented on diagram.
- The “Discussion” section should be supplemented with information about other authors’ investigations on the effect of UV radiation on plant growth and development, if there are any, according to the authors. It is necessary to compare your data with the data presented by other authors.
- - In the section "Materials and methods" it is necessary to specify the extracting agent used for sample preparation, specify the name, brand, country of manufacture of all equipment used in the study.
Reviewer 2 Report
Comments and Suggestions for Authors
Ijms 2811187 review 10.02.2023
1. What is the main question addressed by the research?
The manuscript “Abscisic acid affects phenolic acid content to increase tolerance to UV-B stress in Rhododendron chrysanthum Pall.” presents results of transcriptomics and metabolomics studies focused on phenolic acids metabolism run on a species originating from the Alps, which was exposed to short-term UV-B (two days, 8h/day) and externally applied ABA.
2. Do you consider the topic original or relevant in the field? Does it address a specific gap in the field? 3. What does it add to the subject area compared with other published material?
The manuscript is another in a series of works published by X. Zhou et al. [1-4] regarding tolerance mechanisms induced in Rhododendron chrysanthum Pall plants. Publications [2-4] were published in MDPI journals and are cited in the current manuscript. So, the topic is not so original but can be considered relevant in the field because it adds another piece of news regarding the response to stress in the species studied.
Modifying the stress response depending on ABA is widely researched; for example, ten years ago, MDPI published a similar article [5]. The concept is not new; the history of ABA research dates back to the mid-1960s, nearly 60 years old. Publications on this subject are still being published. They concern more and more species and research methods. The manuscript under review is one such publication, bringing knowledge about Rhododendron chrysanthum.
4. What specific improvements should the authors consider regarding the methodology? What further controls should be considered?
Dear Authors, please re-write the sentence: Selected uniformly grown R. chrysanthum seedlings were treated with radiation, before radiation, 400nm long-pass filters were placed on the culture bottles of group M, and 295nm long-pass filters were placed on the culture bottles of groups N and Q. (lines 483-484). Please write materials in straight-forward way, starting from the sentence:
The experimental material consisted of 8-month-old R. chrysanthum seedlings, which were planted in ¼ MS medium and grown in an artificial climate chamber simulating an alpine environment: temperature 16 - 18℃, 14/10 h of day/night, 60% of relative humidity) for xxx days (how long??) . The light spectrum was: ?????. Please describe further steps: Plants were divided into groups xxx, and subjected to different environmental conditions. Please list the conditions one by one or summarize them in the table.
Other methods are well described and mostly published earlier [1-4]
5. Are the conclusions consistent with the evidence and arguments presented, and do they address the main question posed?
The conclusions suggest that the results presented constitute an entirely new discovery. However, the reviewed manuscript is a repetition and confirmation of previously conducted research, as shown by i.e. [5]. Therefore, the conclusions should be reworded to express that the article is a confirmation of previously carried out research. And that the novelty is related to the species studied and the methods used.
As the Conclusions, the Abstract should also show that the topic has been studied for many years and that the novelty is in the species studied and the methods used.
6. Are the references appropriate?
Yes
7. Please include any additional comments on the tables and figures.
I do not have additional comments.
Literature discussed in the review:
1. Zhou, Xiaofu & Chen, Silin & Wu, Hui & Yang, Yi & Xu, Hongwei. (2017). Biochemical and proteomics analyses of antioxidant enzymes reveal the potential stress tolerance in Rhododendron chrysanthum Pall. Biology Direct. 12, https://doi.org/10.1186/s13062-017-0181-6
2. Sun, Q.; Liu, M.; Cao, K.; Xu, H.; Zhou, X. UV-B Irradiation to Amino Acids and Carbohydrate Metabolism in Rhododendron chrysanthum Leaves by Coupling Deep Transcriptome and Metabolome Analysis. Plants (Basel) 2022, 11, 626. https://doi.org/10.3390/plants11202730
3. Liu, M.; Sun, Q.; Cao, K.; Xu, H.; Zhou, X. Acetylated Proteomics of UV-B Stress-Responsive in Photosystem II of Rhododendron chrysanthum. Cells 2023, 12, https://doi.org/10.3390/cells12030478
4. Sun, Q.; Zhou, X.; Yang, L.; Xu, H.; Zhou, X. Integration of Phosphoproteomics and Transcriptome Studies Reveals ABA Signaling Pathways Regulate UV-B Tolerance in Rhododendron chrysanthum Leaves. Genes (Basel) 2023, 14, https://doi.org/10.3390/genes14061153
5. Ibrahim, M.H.; Jaafar, H.Z.E. Abscisic Acid Induced Changes in Production of Primary and Secondary Metabolites, Photosynthetic Capacity, Antioxidant Capability, Antioxidant Enzymes and Lipoxygenase Inhibitory Activity of Orthosiphon stamineus Benth. Molecules 2013, 18, 7957-7976. https://doi.org/10.3390/molecules18077957
Comments on the Quality of English LanguageIn general, English is understandable.
